# Nebivolol as a Potent TRPM8 Channel Blocker: A Drug-Screening Approach through Automated Patch Clamping and Ligand-Based Virtual Screening

**DOI:** 10.3390/membranes12100954

**Published:** 2022-09-28

**Authors:** Farhad Jahanfar, Laura Sadofsky, Alyn Morice, Massimo D’Amico

**Affiliations:** 1Di.V.A.L. Toscana S.r.l., Via Madonna del Piano 6, 50019 Sesto Fiorentino, Italy; 2Department of Medical Biotechnologies, University of Siena, 53100 Siena, Italy; 3Centre for Atherothrombosis and Metabolic Disease, Hull York Medical School, University of Hull, Hull HU6 7RX, UK; 4Respiratory Research Group, Hull York Medical School, Castle Hill Hospital, Cottingham HU16 5JQ, UK

**Keywords:** TRPM8, repurposing, nebivolol, carvedilol, automated patch clamp

## Abstract

Transient Receptor Potential Melastatin 8 (TRPM8) from the melastatin TRP channel subfamily is a non-selective Ca^2+^-permeable ion channel with multimodal gating which can be activated by low temperatures and cooling compounds, such as menthol and icilin. Different conditions such as neuropathic pain, cancer, overactive bladder syndrome, migraine, and chronic cough have been linked to the TRPM8 mode of action. Despite the several potent natural and synthetic inhibitors of TRPM8 that have been identified, none of them have been approved for clinical use. The aim of this study was to discover novel blocking TRPM8 agents using automated patch clamp electrophysiology combined with a ligand-based virtual screening based on the SwissSimilarity platform. Among the compounds we have tested, nebivolol and carvedilol exhibited the greatest inhibitory effect, with an IC_50_ of 0.97 ± 0.15 µM and 9.1 ± 0.6 µM, respectively. This study therefore provides possible candidates for future drug repurposing and suggests promising lead compounds for further optimization as inhibitors of the TRPM8 ion channel.

## 1. Introduction

Ion channels are involved in almost any biological process; accordingly, they are significant targets in drug discovery and a cause of concern as off-targets during the risk assessment of new drugs. In fact, together to G-Protein Coupled Receptors (GPCR), ion channels are the most common class of membrane proteins to have a causal relationship with human pathologies, and attempts to find new validated ion-channel drug targets and create new drugs against these have been challenging. One such example is the family of transient receptor potential (TRP) cation channels [1]. In the mammalian genome, there are six different subgroups of TRP channels based on their sequence homology: TRP ankyrin (TRPA), vanilloid (TRPV), canonical (TRPC), mucolipin (TRPML), polycystin (TRPP), and melastatin (TRPM) [2].

In particular, TRPM8 belongs to the melastatin TRP channel subfamily and, being a non-selective Ca^2+^-permeable ion channel with a permeability ratio of PCa^2+^/PNa^+^~3, is a member of the voltage-gated superfamily. TRPM8 has a multimodal gating, which can be activated by low temperatures (<28 °C), depolarization, cooling compounds such as menthol and icilin, and changes in osmolarity [3]. TRPM8 channels are essential for thermoregulation, responding to innocuous cool, cold hypersensitivity, noxious cold, and cooling-mediated analgesia [4], and they are localized in testis and prostate tissues [5], as well as in the bladder and the male urogenital tract [6], the vascular smooth muscle [7], the eye [8], the immune system [9], and the lung [10]. Moreover, TRPM8 is accumulated to high levels in the peripheral nervous system [3]. Growing evidence suggests that this channel plays a role in various diseases, such as neuropathic pain [11], overactive bladder syndrome [12], migraine [13], cancer [14,15], and cough [16]. Accordingly, the DisGeNET database [17] contains 127 diseases associated with the *TRPM8* gene. 

Among the numerous synthetic and natural inhibitors of TRPM8 that have been discovered, only three compounds (AMG-333, PF-05105679, and cannabidivarin) have entered a clinical trial to date. Of these, AMG-333 (NCT01953341) and PF-05105679 (NCT01393652) did not pass phase I, and cannabidivarin (NCT02365610) is currently in phase II. Thus, despite many efforts, there are, as of yet, no inhibitor clinical candidates specific for TRPM8 [18,19]. Given that the drug development and approval process of a new compound takes an estimated average of 8 to 12 years [20], a drug repurposing approach would provide an alternative fast-track solution to this current bottleneck.

Drug repurposing or drug repositioning can be defined as the strategy to search for a new therapeutic use for previously approved drugs [21]. Because ample information about approved drugs exists in terms of pharmacodynamics, pharmacokinetics, and toxicities, this approach is cheaper and faster, posing less of a risk of failure compared with traditional drug discovery [22]. Multitarget drugs are potential candidates for repurposing; their ability to interact with several targets (which is known as polypharmacology) could be both responsible for side effects and essential for the repurposing strategy based on their targets [23]. Furthermore, their structures could be considered as lead compounds for drug discovery [24].

In this study, we sought to identify TRPM8 inhibitors by combining the methods of electrophysiology and ligand-based virtual screening. To optimize our search of potent drugs for screening, we have used a ligand-based virtual screening web tool [25,26], which has been performed to generate virtual drug candidates on the TRPM8 ion channels, and the automated patch clamp device IonFlux 16, which is routinely used to screen and evaluate the drugs’ effects on the ion channels [27].

Our initial group of compounds included sodium channel blockers due to their diverse structures [28] and because they have been successfully repurposed for several diseases [29], some of which were effective against col allodynia [30].

In total, we identified four inhibitors of TRPM8 channels: propafenone, propranolol, carvedilol, and nebivolol. Of these, the last two drugs have showed the greatest inhibitory effects.

The results of this study should provide additional candidates for drug repurposing, including promising lead compounds capable of inhibiting TRPM8 channels.

## 2. Materials and Methods

### 2.1. Cell Culture and Preparation

HEK293 cells, stably expressing human TRPM8 (GenBank number NM_024080) [31], were cultured in T75 tissue culture flasks using Dulbecco’s modified Eagle medium (Euroclone, Pero, Italy), supplemented with 10% FBS (Euroclone, Pero, Italy) plus Geneticin (Sigma-Aldrich, St. Louis, MI, USA) 800 μg/mL at 37° C, in a humidified 5% CO_2_ atmosphere. The cells were maintained until they reached 70~80% confluency. Prior to use, the cells were washed twice with Ca^2+^ and Mg^2+^ free PBS (Euroclone, Pero, Italy), subsequently detached using Accutase (Euroclone, Pero, Italy), and kept in suspension using CHO-SFM-II medium (Gibco, by Thermo Fisher Scientific, Carlsbad, CA, USA). The cells were then resuspended in the external solution after a centrifugation at 1000 rpm for 2 min. Single-cells suspension, with a final concentration of 3–5 million cells/mL, was required for the automated patch clamp electrophysiology procedure.

### 2.2. Automated Patch Clamp Electrophysiology

We performed automated patch clamp recordings by using the IonFlux 16 system (Fluxion Biosciences, Alameda, CA, USA). The procedure includes four automated phases—prime, trap, break, and data acquisition—in the whole-cells voltage clamp [27]. The cells were voltage clamped at a holding potential of −60 mV, stepped to −100 mV, ramped to +100 mV in 200 ms, and finally stepped back to −60 mV. Voltage command was executed every 5 s, and currents were recorded with a sampling rate of 5 kHz. The cells were activated by the continuous perfusion of 300 µM menthol for 30 s and then exposed to drugs at different concentrations, co-applied with 300 µM menthol for 90 s. The interval range between each application was 90 s, and intervals included washing cells with an external solution-containing vehicle. However, the duration of each application was adjusted based on each drug’s lipophilicity and kinetic of inhibition. For leakage subtraction, leakage was measured by applying a short 20 mV pulse (typically from −80 mV to −100 mV) at the beginning of each sweep to measure the current difference. By assuming linear leakage and subtracting proportional currents from each of the different voltage command segments, the software performs automatic leakage subtraction. At the end of each experiment, a full block solution of *N*-(3-Aminopropyl)-2-[(3-methylphenyl) methoxy]-*N*-(2-thienylmethyl)benzamide hydrochloride (AMTB) 10 µM was applied to reach 100% inhibition. The extracellular solution contained 140 mM NaCl, 4 mM KCl, 1 mM MgCl_2_, 10 mM glucose, and 10 mM HEPES, pH 7.4 with NaOH and 310 mOsm. We removed Ca^2+^ from the external solution to avoid the Ca^2+^-dependent desensitization of TRPM8. The intracellular solution contained the following (mM): 60 CsCl, 70 CsF, 5 Na_2_ATP, 10 HEPES, and 10 EGTA, pH 7.2 with CSOH and 295 mOsm. All recordings were performed at a temperature of 20 ± 1 °C.

### 2.3. Buffers and Compounds

All compounds were obtained from Sigma-Aldrich. Stock solutions were prepared in 50 mM of dimethyl sulfoxide (DMSO) and stored at −20° C. Serial dilutions of working compounds were prepared freshly by adding a proper amount to the extracellular solution containing menthol, keeping the concentration of DMSO at 0.3%.

### 2.4. Ligand-Based Virtual Screening

We used the SwissSimilarity web tool (http://www.swisssimilarity.ch accessed on 20 January 2022) [25] to identify ligands among the approved drug molecules with a similar structure to the most potent drug we found in the first step of our procedure. The ligand-based virtual screening was performed using different approaches such as FP2 fingerprints, spectrophores, electroshape, and align-IT screenings [26].

### 2.5. Molecular Docking Study

To predict and analyze possible protein–ligand interactions with the known binding site on TRPM8, we performed molecular docking studies using Autodock vina 1.1.2 software (Center for Computational Structure Biology, La Jolla, CA, USA) [32]. The TRPM8 structure (PDB code: 6O6R) [33] was obtained from the RCSB Protein Data Bank (available at http://www.rcsb.org accessed on 20 January 2022, Research Collaboratory for Structural Bioinformatics (RCSB)). PDB files for ligands were obtained from DrugBank [34] (https://go.drugbank.com/ accessed on 20 January 2022), and AutoDock tool 1.5.7 software (Center for Computational Structure Biology, La Jolla, CA, USA) was used to prepare the structure. All non-protein molecules were removed, polar hydrogen atoms were added, and non-polar hydrogen atoms were merged. We defined flexible residues around the binding pocket, and the grid dimension and box were fitted on the binding site. The structures of proteins and ligands were converted to pdbqt format. UCSF-Chimaera software (Resource for Biocomputing, Visualization, and Informatics (RBVI), San Francisco, CA, USA) was used to perform the analyses of all docked poses [35], and poses with a minimum binding energy were selected for analysis. The 2D representation of ligand-protein binding was generated using the program LigPlot^+^ (LigPlot^+^ v.2.2, EMBL’s European Bioinformatics Institute (EMBL-EBI, Hinxton, Cambridgeshire, UK) [36].

### 2.6. Data Analysis

The analysis of the data was carried out using the ‘data analyzer software’ (Data Analyzer v. 5.0 (beta), Fluxion Biosciences, Alameda, CA, USA), Microsoft Excel 365, and the Prism 9.0 software (GraphPad Software Inc., San Diego, CA, USA). The average current density of the last three recordings at +80 mV in each application period was used as the current amplitude. The inhibition of drugs on menthol-induced currents was normalized using the current amplitude obtained using 300 μM menthol. Menthol-independent leak currents were subtracted from the total currents in the presence of 300 μM menthol. The percentage inhibition of the TRPM8 channel current at each compound concentration was determined using the following equation:$$\%\;Inhibition\;of\;I=100\times \left(\frac{I_{control}-I_{drug}}{I_{control}}\right)$$

IC_50_ and EC_50_ were calculated by fitting the four-parameter Hill equation using GraphPad Prism.

Four-parameter Hill equation:$$f\left(x\right)=I_{min}+\left(\frac{I_{max}-I_{min}}{1+{\left(\frac{C_{50}}{x}\right)}^{h}}\right)$$

In the four-parameter logistic equation, *I_min_* = 0 and *I_max_* = 100 (agonist), or *I_min_* = 100 and *I_max_* = 0 (antagonist); h is the Hill coefficient, C is the concentration of the tested compound, and C_50_ is the EC_50_ (activator) or IC_50_ (inhibitor) values. Data are expressed as the mean ± standard deviation (SD).

## 3. Results

We first studied the activation of TRPM8 by menthol. By applying a concentration of 50 µM, a detectable outward current with a linear increase at a positive potential (ohmic) and a negligible current at a negative potential were evoked. At 300 µM, the current amplitude evoked by menthol appeared relatively stable and showed little or no rundown. Menthol produced a concentration-dependent current, and the effect was reversed by washing out (Figure 1A). As shown in Figure 1C, the relative activation quickly reached a steady state condition. In Figure 1B, using a voltage ramp protocol [37], the relationship between the current response and menthol concentration is shown, with an EC_50_ of 118.2 ± 14 µM. The menthol-activated currents were fully blocked after the application of 1 µM AMTB, which had an IC_50_ of 355.5 ± 25 nM (Figure 1D).

To identify novel TRPM8 inhibitors, we next screened a set of drugs, including the sodium ion channel blockers: mexiletine, propafenone, ranolazine, carbamazepine, tetracaine, flecainide, rufinamide, gabapentin, oxcarbazepine, lamotrigine, phenytoin, and riluzole. None of these drugs had activating effects, and most of them did not provide any blocking effect on the outward TRPM8 currents induced by menthol. Only flecainide, phenytoin, and propafenone showed a blocking effect, though, notably, a partial block was only obtained with flecainide and phenytoin at concentrations higher than 50 µM. Only propafenone had a significant blocking capacity below 50 µM (Table 1).

The IC_50_ for this compound was 16.8 ± 5 µM (Figure 2A). Hence, to optimize our screening for potent TRPM8 inhibitors, we considered the structure of propafenone as a hit structure.

We performed a ligand-based virtual screening using the SwissSimilarity platform. Our virtual screening library, comprising only FDA (Food and Drug Administration)-approved drugs, compared 1516 compounds using different methods: FP2 fingerprint, electroshape, spectrophores, and combined methods [26]. The results obtained by these approaches allowed us to identify 11,215, 400, and 215 candidates as possible TRPM8 inhibitors, respectively. Most of the drugs with the highest scores in terms of similarity to propafenone were categorized as β-blockers. Based on these results, we selected seven β-blockers, including: propranolol, atenolol, metoprolol, carvedilol, nadolol, nebivolol, and acebutolol. Of these, propranolol, carvedilol, and nebivolol showed a blocking effect on TRPM8 menthol-evoked currents (Table 2).

In particular, propranolol had an IC_50_ higher than propafenone (45 ± 7 µM vs. 16.8 ± 7 µM, Figure 2A,B). Carvedilol, with an IC_50_ = 9.1 ± 0.6 µM, results a more effective blocker than propafenone (Figure 2C). Nebivolol, with an IC_50_ = 0.97± 0.15 µM, turned out to be the most potent blocker we screened (Figure 2D). Moreover, in comparison to propafenone and propranolol, nebivolol and carvedilol demonstrated an irreversible inhibiting effect at concentrations higher than IC_50_, with little to no current recovery, even after a prolonged washing (data not shown).

Molecular docking studies, using Autodock vina 1.1.2 software, allowed us to predict the best binding affinities (interactions) between the four inhibiting β-blockers and the S1, S2, S3, and S4 transmembrane domains of the TRPM8 channel (Figure 3A,B).

The minimum binding energies for nebivolol, carvedilol, propafenone, and propranolol were −11.4, −13.3, −9.1, and −10.4 kcal/mol, respectively. As shown in the 2D plot, there are five different potential hydrogen-bond interactions between nebivolol and the TRPM8 residues Gln776, Asn790, Glu773 (double interaction), and Asp793. Additionally, two π–π stacking interactions were identifiable at the binding site level: a face-to-face interaction with Tyr736 and a T-shaped interaction with Tyr995. Other residues (Phe729, Arg998, Phe1003, Val733, Ile836, Arg832) are predicted to be involved in hydrophobic interactions (Figure 3C). By contrast, carvedilol contains one hydrogen bond site with Ser840 and two face-to-face π–π stacking interactions at Phe729 and Tyr736 (Figure 4A). Propafenone has two hydrogen bond sites at Asn732 and Asp772 and one face-to-face π–π stacking interaction at Tyr736 (Figure 4B). Finally, propranolol has no hydrogen bonds and only one face-to-face π–π stacking interaction at Tyr736 (Figure 4C).

## 4. Discussion

In this study, we have developed an automatic patch clamp electrophysiology assay based on the IonFlux 16 system for the screening of potential TRPM8 inhibitors. We were able to obtain long-term recordings and stable currents from stably transfected HEK-293 cell lines. There are specific differential technical tips to be considered when performing automated versus manual patch clamping, such as the homogeneous and high-level expression of target ion channels for the robust recording of currents, the size of cells in single-cell suspensions for efficient cell trapping, the quality of cell membranes for easier sealing, and the whole-cell configuration for long-term and stable recording.

In line with previous reports [38,39,40], we found an EC_50_ for the TRPM8 activator menthol of 118.2 ± 14 µM and an IC_50_ for the well-known TRPM8 inhibitor AMTB of 355.5 ± 25 nM.

Moreover, by ligand-based virtual screening using the SwissSimilarity platform, and taking into consideration the broad range of FDA-approved drugs capable of binding different pharmacological targets, we identified four inhibitors of the TRPM8 ion channels. Among the newly identified TRPM8 blockers, nebivolol was identified as the most potent inhibitor, with an IC_50_ at submicromolar concentrations [41]. This drug belongs to the third generation of β-adrenergic receptor antagonists, and it is highly selective to the β1-adrenergic receptor [42]. Additionally, it exhibits a nitric oxide-mediated vasodilation by stimulating the β3-adrenergic receptor [43].

Thus, nebivolol may represent a promising lead compound for further evaluation and modification. For instance, it could be used as a starting structure for drug discovery—an approach that increases the likelihood of identifying new compounds with drug-like properties.

This type of identification strategy, which is based on the known side effects of available drugs rather than on the principle effects, is termed the Selective Optimization of Side Activities (SOSA) approach [24]. The results of this research further shed additional light on the polypharmacology of identified drugs. The ability of a drug to have several targets could be advantageous in treating complex diseases [44], and it could explain the side effects. To this end, it is worth noting that, in the recent screening studies [45,46], nebivolol was also identified as a potential candidate for repurposing. These findings, along with our study, reveal a complex polypharmacology of nebivolol.

Molecular docking studies allowed us to unravel the possible interaction between TRPM8 and the newly identified blockers at the transmembrane level. Such binding sites are known as ligand hubs for both agonists and antagonists [47]. According to our docking analysis, nebivolol possesses five possible hydrogen bonds, while two π–π stacking connections stabilize the complex. Further, additional hydrophobic interactions at other sites contribute to the enhanced affinity of nebivolol to the TRPM8 binding site.

These interactions likely induce conformational rearrangements and result in TRPM8 blocking or in the inhibition of a menthol-dependent activation. In comparison to other drugs, nebivolol is capable of more non-covalent interactions with the TRPM8 binding site, as evidenced by IC_50_ values.

Interestingly, the results of the Autodock vina software showed that carvedilol had a lower binding energy (−13.3 kcal/mol) in comparison to nebivolol (−11.4 kcal/mol). Such outcomes disagree with our electrophysiological experimental data, which showed, on the other hand, that the inhibitory effect of nebivolol turned out to be ten times stronger. The inconsistency between the binding energy and the pharmacology effect was reported for Autodock vina, and it was mentioned that Autodock vina adopts very accurate binding poses [48].

Another possible mechanism for the blocking of TRPM8 ion channels might be ascribed to membrane lipid bilayer-drug interactions. Amphiphilic drugs with both hydrophilic and lipophilic properties could alter the function of many diverse transmembrane proteins, such as ion channels. Transmembrane proteins are energetically coupled to the transmembrane lipid bilayer; thus, amphiphilic drugs such as propafenone, propranolol, carvedilol, and nebivolol could change their functions by partitioning into the cell membrane and altering the bilayer contribution to energetics and conformational changes [48]. Such a mechanism was reported for a variety of drugs, as well as propranolol. Propranolol interacts with the phospholipid bilayer in a concentration-dependent manner and thus could alter the interactions between ion channels and the lipid bilayer [49].

Interestingly, TRPM8 is known to be particularly susceptible to drug-induced membrane alterations since it has several interaction sites for the membrane lipid bilayer. For instance, TRPM8 is localized in cholesterol-rich membrane domains, and a decrease in cholesterol disrupts the lipid rafts, which causes a significant increase in the TRPM8 response to cold and menthol [50].

Similarly, a known key regulator of TRPM8, phospholipid phosphatidylinositol 4,5-bisphosphate (PIP2), activates TRPM8 channels even in the absence of chemical or thermal stimulation, and it also enhances TRPM8 sensitivity to cold and menthol [51]. By contrast, ethanol confers an inhibitory effect that has been ascribed to its capacity to alter the PIP2–TRPM8 interaction [52]. Moreover, it has been reported that carvedilol is able to inhibit Kir2.3, using the same mode of action [53]. We postulate that this might also be the mechanism responsible for the inhibitory effect of carvedilol on TRPM8 ion channels.

Interestingly, such an effect has been reported for propafenone in blocking cardiac Kir2.x channels: by reducing the negative charges in their cytoplasmic pore, the drug is able to reduce the affinity of those ion channels to PIP2 [54]. In this way, the same mechanism might also be induced by propafenone in blocking TRPM8 channels.

Although we cannot exclude the possibility that the identified drugs interact directly with TRPM8 ion channels, such drug–membrane interplay could, however, explain the complex polypharmacology or promiscuity characterizing these molecules [55].

## 5. Conclusions

In summary, we have developed an assay for the rapid identification of TRPM8 channel inhibitors via automated patch clamp electrophysiology combined with virtual screening. Among a variety of FDA-approved drugs, we identified four molecules, of which nebivolol proved to be a novel potent TRPM8 inhibitor. Hence, we propose nebivolol as a promising structure for further modification in order to develop new TRPM8 blockers. Moreover, our approach could facilitate the identification of other new validated ion-channel drug targets and help develop new drugs, since it can be applied to any ion channel and transporter. Importantly, by exploiting drug repurposing, this approach could be a valid strategy for use in a wide range of pathologies.

## Figures and Tables

**Figure 1 membranes-12-00954-f001:**
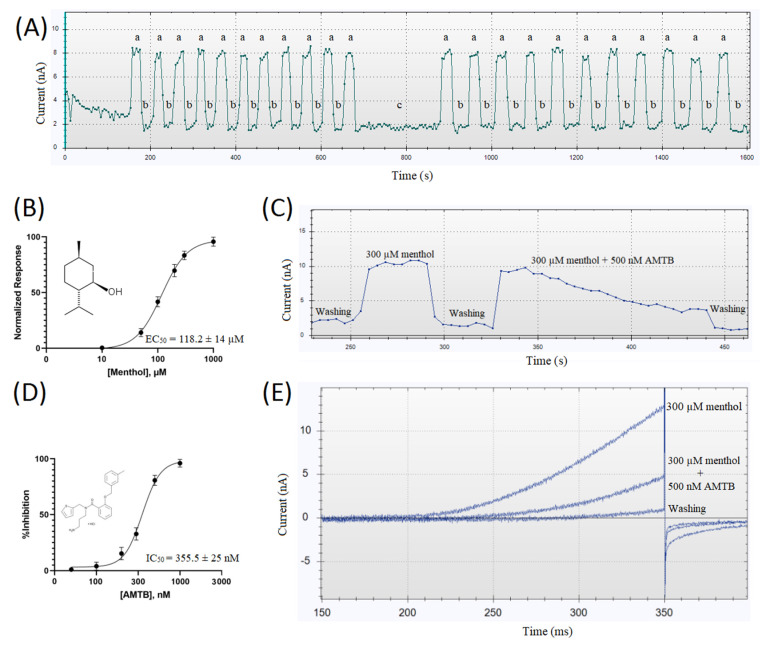
(**A**) Recorded outward currents at +80 mV from HEK-293 cells stably expressing TRPM8 were obtained using the Ionflux 16 device. Currents were evoked by a voltage ramp from −100 to +100 mV, in the presence of 300 µM menthol (a) or 0.3% DMSO (vehicle) (b), and then (c) washed with 0.3% DMSO for a longer period (200 s). Note that menthol evokes a reproducible and time-independent current. (**B**) Dose response curve for menthol with an EC_50_ = 118.2 ± 14 µM. (**C**) Outward currents recorded at +80 mV from HEK-293 cells stably expressing TRPM8, during washing with an external solution containing the vehicle followed by perfusion with 300 µM menthol alone or in combination with 500 nM AMTB. (**D**) Dose response curve for AMTB with an IC_50_ = 355.5 ± 25 nM; IC_50_ and EC_50_ were calculated by fitting the four-parameter Hill equation. All data are shown as the mean ± SD (*n* ≥ 6). (**E**) Outward currents elicited in HEK-293 cells stably expressing TRPM8 by 300 µM menthol alone or in combination with 500 nM AMTB and after washing with an external solution containing the vehicle, using a voltage ramp protocol from −100 to +100 mV over 200 ms.

**Figure 2 membranes-12-00954-f002:**
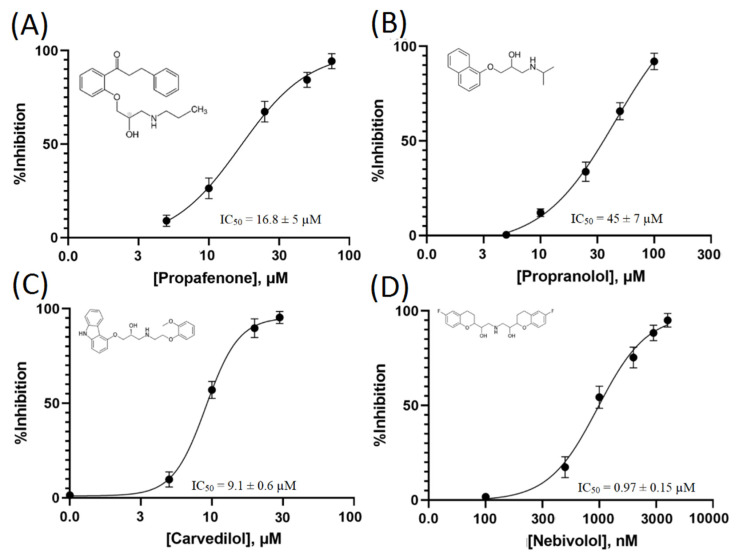
Dose response curves of the selected β-blockers capable of inhibiting TRPM8 channels. Data were obtained using the IonFlux16 device. (**A**) Propafenone, (**B**) Propranolol, (**C**) Carvedilol, (**D**) Nebivolol. All IC_50_ were calculated by fitting the four-parameter Hill equation; all data are shown as the mean ± SD (*n* ≥ 6).

**Figure 3 membranes-12-00954-f003:**
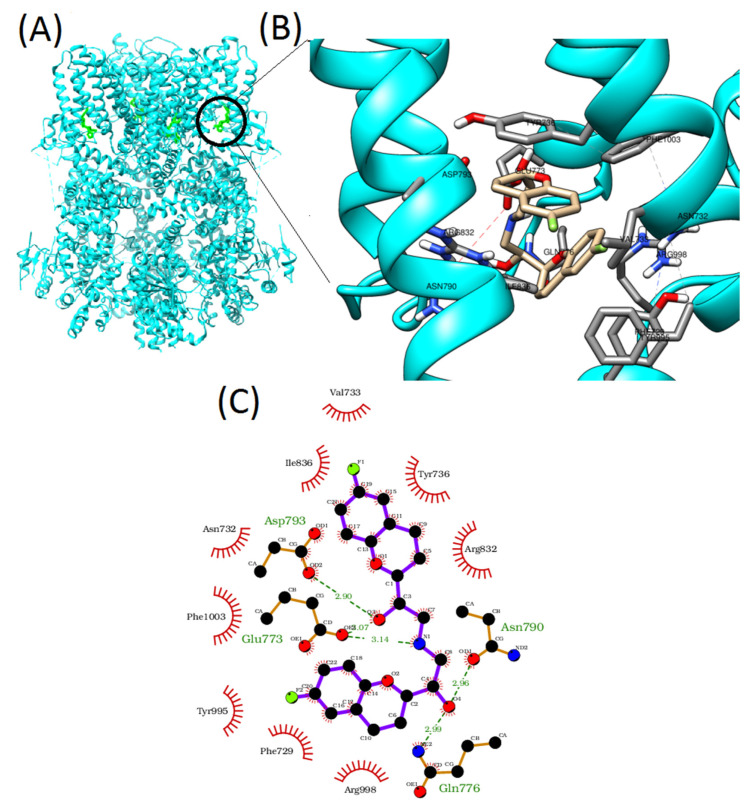
(**A**) Structure of the TRPM8 protein showing the four interaction binding sites, highlighted in green. (**B**) 3D depiction of nebivolol-TRPM8 binding. (**C**) 2D depiction of the intermolecular interactions between TRPM8 and nebivolol. The 2D plots were generated by LigPlot^+^. Hydrogen bonds are shown as green dotted lines, while the red spoked arcs represent hydrophobic contacts of residues with each drug. In the represented structures, carbon, oxygen, flour, and nitrogen are in black, red, green, and blue, respectively.

**Figure 4 membranes-12-00954-f004:**
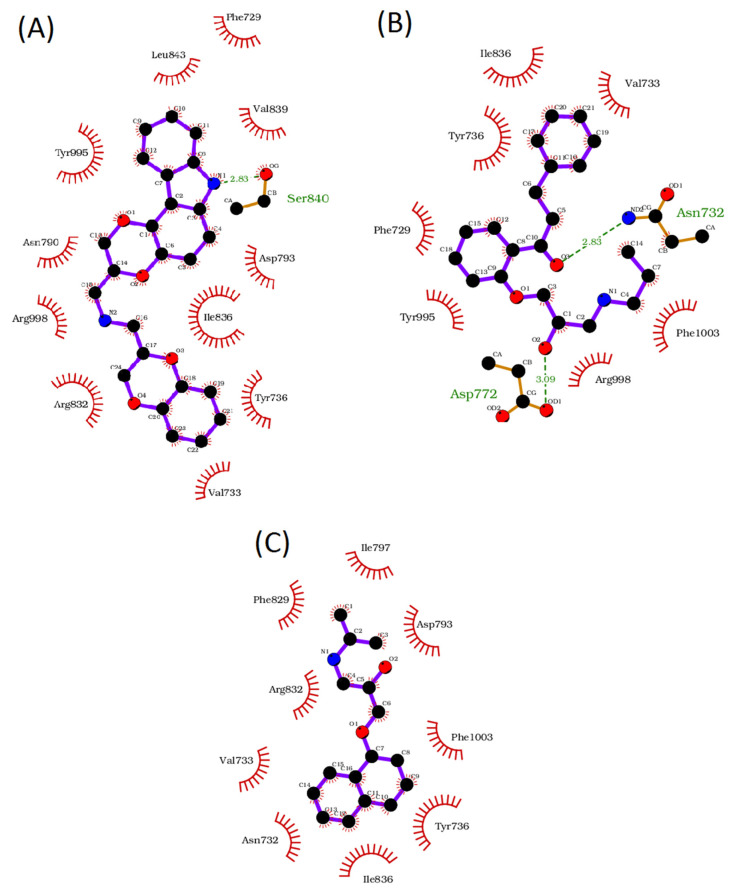
2D depiction of the intermolecular interactions between TRPM8 and carvedilol (**A**), propafenone (**B**), and propranolol (**C**). The 2D plots were generated by LigPlot^+^. Hydrogen bonds are shown as green dotted lines, while the red spoked arches represent hydrophobic contacts of residues with each drug. In the represented structures, carbon, oxygen, flour, and nitrogen are in black, red, green, and blue, respectively.

**Table 1 membranes-12-00954-t001:** TRPM8 percent inhibition data.

Drug Name	Average Percentage Inhibition at 10 µM	
Propafenone	26	(*n* = 7)
Flecainide	0	(*n* = 6)
Phenytoin	0	(*n* = 6)
Oxcarbazepine	0	(*n* = 6)
Lamotrigine	0	(*n* = 6)
Riluzole	0	(*n* = 6)
Carbamazepine	0	(*n* = 6)
Gabapentin	0	(*n* = 6)
Rufinamide	0	(*n* = 6)
Ranolazine	0	(*n* = 6)
Tetracaine	0	(*n* = 6)
Mexiletine	0	(*n* = 6)

**Table 2 membranes-12-00954-t002:** TRPM8 percent inhibition data.

Drug Name	Average Percentage Inhibition at 10 μM	
Nebivolol	100	(*n* = 8)
Carvedilol	57	(*n* = 7)
Propranolol	12	(*n* = 6)
Metoprolol	0	(*n* = 6)
Atenolol	0	(*n* = 6)
Nadolol	0	(*n* = 6)
Acebutolol	0	(*n* = 6)

## Data Availability

All the data is available on demand.

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
