# Peer review of "Nebivolol as a Potent TRPM8 Channel Blocker: A Drug-Screening Approach through Automated Patch Clamping and Ligand-Based Virtual Screening"

_membranes, 2022, doi:10.3390/membranes12100954_

Round 1
Reviewer 1 Report
This is a small but interesting study aimed at identifying new blockers of the TRPM8 channel. Using a combination of electrophysiology and virtual screening the authors report nebivolol has TRPM8 blocking effects with an IC50 ~ 1µM. The approach and data seem on the whole solid and will be of interest to readers.
My major comments are:
- The manuscript is not well written with numerous grammatical errors and errors in sentence construction which combine to create confusion for the reader. This needs to be corrected throughout.
- There is no rationale or discussion as to the selection of compounds for the original electrophysiology screen - why sodium channel blockers ? If a larger set was screened this should be defined in terms of its content and ideally the data shown from the whole screen (inactives can sometimes be as insightful as actives).
- The authors suggest in the conclusion that nebivolol could be used as a TRPM8 therapeutic - bearing in mind its activity at beta1 adrenoceptors is <10nM I don't feel this supported. It may represent a novel scaffold for further development so I feel this sentence should be softened.
- In relation to point 4, nebivolol has come up as a hit in multiple repurposing screens recently (eg. Yang et al 2022 Bioorg Med Chem Lett 55, 128445; Anand et al 2021 PLoS One 16(3):e0248553) - these should be mentioned in the discussion in relation to polypharmacology.
Minor comments:
- Introduction, line 37; the term ‘off targets’ is not one commonly used and is confusing for the reader. I would suggest that ‘selectivity concerns’ or a similar phrase is more appropriate.
-
Line 40,41: suggesting that drug discovery efforts against TRP channels have failed is perhaps controversial, as capsaicin and menthol are in clinical use. Perhaps ‘challenging’ would be better ?
Reviewer 2 Report
The current manuscript, titled” Nebivolol as a potent TRPM8 channel blocker: a drug screening 2 approach through automated patch clamping and ligand based virtual screening”, used auto-patch clamp to screen the FDA approved drugs and found that Nebivolol has potential inhibition for TRPM8 currents. In my opinion, the content of this manuscript is not enough for publication in the journal. They used HEK293-TRPM8 cell line. It is not clear that HEK cell expressed with human TRPM8 or rodent TRPM8. The detail information of this cell line should add in Materials and Methods. Author described that they applied NMDG solution to indicate the leak current. However, I did not find the record with NMDG. Then, it is hard to exclude the leak current. Authors collected the current amplitudes at +80 mV, the outward currents. How about the inward current? I am not sure how the IC50 obtained. Do the inhibition of different concentration compound obtain from one cell or different cells? Figure 2 showed the IC50s of four compounds. How about their maximal inhibition? A table or figure is needed here to illustrate the maximal inhibition. The resolution of figure 3 is not high enough.
Reviewer 3 Report
In the present study, the authors developed an assay for rapid identification of TRPM8 channel inhibitors by using automated whole-cell patch-clamp recording combined with virtual screening. Among a variety of FDA-approved drugs, they identified four molecules; in particular, nebivolol as a novel potent TRPM8 inhibitor. Therefore, they suggest that TRPM8 might be a new nebivolol target, and nebivolol might be a potential therapeutic agent for TRPM8 associated diseases. The manuscript was well written and the conclusion was supported by the experimental data. I only have minor concerns for authors to check the typo error and reference format throughout the manuscript.
- Line 88, "CO2" should be changed into "CO2".
- For references cited in the lists, the authors should follow the guideline for submission.
Round 2
Reviewer 1 Report
Comments and concerns raised in the original review have been addressed.
Author Response
We are grateful for your consideration of our manuscript and we thank you for your careful evaluation, which has helped us improve the manuscript.
Reviewer 2 Report
1. Please provide the GenBank number of TRPM8ï¼›
2. For TRP currents, the inhibition for inwards currents has more physiological significance. Author may need to verify if the compound has similar inhibition for inward current of TRPM8 by using extracellular solution with Calcium.
